# Exploring the Sensitivity of Prodromal Dementia with Lewy Bodies Research Criteria

**DOI:** 10.3390/brainsci12121594

**Published:** 2022-11-22

**Authors:** Joseph R. Phillips, Elie Matar, Kaylena A. Ehgoetz Martens, Ahmed A. Moustafa, Glenda M. Halliday, Simon J. G. Lewis

**Affiliations:** 1Faculty of Medicine and Health, Brain and Mind Centre, School of Medical Sciences, University of Sydney, Sydney, NSW 2050, Australia; 2School of Psychology & Marcs Institute for Brain and Behaviour, Western Sydney University, Sydney, NSW 2145, Australia; 3Department of Kinesiology, Faculty of Health, University of Waterloo, Waterloo, ON N2L 3G1, Canada; 4Department of Human Anatomy and Physiology, The Faculty of Health Sciences, University of Johannesburg, Johannesburg 2092, South Africa; 5School of Psychology, Faculty of Society and Design, Bond University, Gold Coast, QLD 4217, Australia; 6Dementia and Movement Disorders Laboratory, Brain and Mind Centre, University of Sydney, Sydney, NSW 2050, Australia

**Keywords:** dementia with Lewy bodies (DLB), prodromal, diagnostic criteria, visual hallucinations, visuospatial, attention, isolated REM sleep behaviour disorder (iRBD)

## Abstract

Dementia with Lewy bodies (DLB) is an insidious neurodegenerative disease characterised by a precipitous decline in cognition, sleep disturbances, motor impairment and psychiatric features. Recently, criteria for prodromal DLB (pDLB) including clinical features and biomarkers have been put forward to aid the classification and research of this ambiguous cohort of patients. Researchers can use these criteria to classify patients with mild cognitive impairment (MCI) with Lewy bodies (MCI-LB) as either possible (either one core clinical feature or one biomarker are present) or probable pDLB (at least two core clinical features, or one core clinical feature and at least one biomarker present). However, as isolated REM sleep behaviour disorder (iRBD) confirmed with polysomnography (PSG) can be included as both a clinical and a biomarker feature, potentially reducing the specificity of these diagnostic criteria. To address this issue, the current study classified a cohort of 47 PSG-confirmed iRBD patients as probable prodromal DLB only in the presence of an additional core feature or if there was an additional non-PSG biomarker. Thirteen iRBD patients demonstrated MCI (iRBD-MCI). In the iRBD-MCI group, one presented with parkinsonism and was thus classified as probable pDLB, whilst the remaining 12 were classified as only possible pDLB. All patients performed three tasks designed to measure attentional deficits, visual hallucinations and visuospatial impairment. Patients also attended clinical follow-ups to monitor for transition to DLB or another synucleinopathy. Findings indicated that the only patient categorised by virtue of having two core clinical features as probable pDLB transitioned over 28 months to a diagnosis of DLB. The performance of this probable pDLB patient was also ranked second-highest for their hallucinatory behaviours and had comparatively lower visuospatial accuracy. These findings highlight the need for more stringent diagnostic research criteria for pDLB, given that only one of the 13 patients who would have satisfied the current guidelines for probable pDLB transitioned to DLB after two years and was indeed the patient with two orthogonal core clinical features.

## 1. Introduction

Dementia with Lewy bodies (DLB) is a common neurodegenerative disease [1,2,3,4] that is characterised pathologically by the presence of Lewy body (LB) inclusions, which are composed of abnormally configured alpha-synuclein protein [5,6,7,8]. Clinically, patients experience a precipitous decline in cognition with significant deficits commonly occurring over just 12 months from onset [9]. Diagnosis of DLB is complex, partly due to criteria that constantly evolve as more is understood about the disease with the latest criteria requiring both clinical biological markers to be present for a diagnosis [10].

Whilst DLB follows an aggressive trajectory, the initial stages might be more subtle and may present over the preceding decades [11,12]. This stage of the disease is termed *prodromal* DLB (pDLB). During the prodromal stage, LBs may not be as profuse, resulting in sub-clinical symptoms that may not be recognised by the patient or their physician [13]. Due to the rapid decline in DLB, early detection and intervention would potentially improve quality of life and allow for the earlier use of future disease modifying strategies.

Genetic testing would represent one mechanism for identifying those people who are at risk of developing DLB, since the presence of a *GBA1* mutation has been identified as a clear risk factor for DLB [14,15], yet most patients do not carry this mutation [16,17]. Recently, increasing evidence has demonstrated that the emergence of isolated rapid eye movement (REM) sleep behaviour disorder (iRBD) represents the strongest predictor for the future development of a synucleinopathy [18,19,20,21]. Rapid eye movement (REM) sleep behaviour disorder is characterised by the loss of skeletal atonia (muscle paralysis) during the REM stage of sleep [22,23]. This loss of atonia releases motor movements that are normally suppressed during REM cycles [24], which results in the dream enactment that is reported clinically. Dream enactment movements can range from minor jerks to complex behaviours (e.g., walking motions, grabbing, talking) [25,26]. These movements may result in the patient falling from bed or causing injury to themselves or their bed partner [25,26]. Patients with iRBD often do not present with other clinical symptoms that would suggest an underlying neurodegenerative disease [27]. They are also likely to continue their day-to-day behaviour as normal. However, recent studies have demonstrated a high number of iRBD patients are likely to develop PD or DLB over their lifetime [13,20,21,28,29].

These studies have highlighted that over 80% of iRBD patients go on to develop Parkinson’s disease (PD) or DLB in roughly equal proportions with a much smaller number developing multiple system atrophy [21,29]. One recent international, multi-centre study evaluating over 1300 iRBD patients has reported a conversion rate to synucleinopathy of 6.3% per year, with 73.5% of cases converting at 12 years following an initial diagnosis [20]. This study also confirmed several clinical features that predicted earlier conversion including abnormal motor testing (especially on objective measures), mild cognitive impairment (MCI), anosmia, impairment on colour vision discrimination, erectile dysfunction, constipation, abnormal dopamine imaging (e.g., DaTSCAN) and the neurophysiological correlate of RBD, namely REM sleep without atonia (RSWA) [20]. However, it should also be noted that 27% of patients who had negative results on these assessments still transitioned to either DLB or PD after 8 years. This study also found that MCI was the only feature that differentiated those iRBD patients who were more likely to transition to DLB rather than PD [20].

Given these greater insights, diagnostic criteria for prodromal DLB have been proposed for use in the research setting [28]. These prodromal DLB guidelines formally identify three specific at-risk patient groups—namely, those with an MCI-onset, delirium-onset or psychiatric-onset presentation. Due to the absence of any detailed, prospective phenotype data, these guidelines offer only limited advice regarding the delirium/psychiatric-onset prodromes [28]. However, characterisation of the MCI-onset sub-group is more aligned with the established clinical criteria for DLB. The essential feature for the diagnosis of MCI-onset prodromal DLB (MCI-LB) is the presence of MCI, defined as exhibiting cognitive impairment across at least one cognitive domain that does not interfere with the patient’s ability to complete everyday activities [30]. The MCI-LB research criteria also evaluates four ‘core’ clinical features: cognitive fluctuations (CF)—characterised by a cycling change between high and low cognition and alertness that affects responsiveness, speech, memory and behaviour [31,32,33], spontaneous visual hallucinations (VH)—defined as percepts (in this case visual) that are not attached to any object [34], iRBD and parkinsonian motor impairments—stiffness or slowing of limb movement, synonymous to the characteristic movements present in PD [35]. Each of these features must not fulfil existing diagnostic criteria to achieve a diagnosis of PD [36]. Three objective biomarkers, abnormal DaTSCAN, REM sleep without atonia (RSWA) as confirmed by video polysomnography (vPSG)—the combination of electroencephalography, electromyography and video footage during the patients’ sleep cycle)—and an abnormal meta-iodobenzylguanidine (MIBG) myocardial scintigraphy scan—which measures the integrity of cardiac norepinephrine receptors, which are impaired in DLB patients [37,38,39,40]—are also considered supportive in the diagnostic criteria [28]. These research criteria also propose two levels of certainty for the diagnosis of MCI-LB, namely *probable* or *possible*. Probable MCI-LB requires either the presence of two or more of the core clinical features or one core feature and one or more objective biomarkers. A research diagnosis of possible MCI-LB requires only one core feature or the presence of only one or more biomarker [28]. 

Whilst the proposed research criteria provide a means to classify this difficult population, the definition of probable MCI-LB may present an issue with ‘over-counting’ in relation to dream enactment (clinical RBD) and its associated biomarker (RSWA). To be confident about the diagnosis of RBD requires confirmation on polysomnography (ideally with video) [22], which should, by definition, be accompanied by the neurophysiological appearance of RSWA [41]. Thus, under the existing guidelines, the presence of RBD confirmed on vPSG in patients with MCI would equate to probable MCI-LB. Clearly, these guidelines might have more value in MCI patients that demonstrate RSWA on vPSG but lack a clinical history of dream enactment. Therefore, in iRBD-MCI patients, it may be desirable to seek an orthogonal biomarker, such as an abnormal MIBG DaTSCAN. Indeed, it should also be recognised that MCI patients have been shown to have a high prevalence of RBD on vPSG (35%) [42]. Moreover, 89% of these iRBD-MCI patients had a non-amnestic cognitive profile that would be consistent with prodromal DLB [28].

Clearly, further studies are needed to confirm the utility of the proposed MCI-LB criteria and one approach would be to determine if behavioural paradigms that have been validated in DLB patients would show any predictive capacity in prodromal cases. The main aim of the current study is to investigate whether the current diagnostic criteria for pDLB as outlined in McKeith et al. [28] or a more conservative criterial that requires a biomarker orthogonal to iRBD, such as MIBG or DaTSCAN, are better suited for identifying pDLB patients. To answer this question, the current study employed both longitudinal and cross-sectional approaches. Patients with iRBD underwent detailed longitudinal assessment at the ForeFront Parkinson’s Disease Research Clinic at the Brain and Mind Centre, University of Sydney. They also completed the Sustained Attention Response task (SART), which has been found to be sensitive to CF in DLB patients [43], the Mental Rotation (MR) task, a visuospatial task that has also been shown to be impaired in DLB patients [44] and the Bistable Percept Paradigm (BPP) which is sensitive to VF in PD [45] and DLB patients [46]. These three tasks were chosen for two reasons: previous studies they have shown them to be sensitive to symptoms (CF, VF and visuospatial impairment) in DLB patients. These tasks also have a range of difficulty, allowing them to be sensitive to features of VF, CF and visuospatial impairment that may not otherwise be detected in a clinical assessment [43,44,46].

It was predicted that the iRBD patients enrolled in this study would include both those with and without MCI. All iRBD patients had their diagnosis confirmed on vPSG, which demonstrated neurophysiological RSWA. Thus, using the research criteria for MCI-LB, those iRBD patients with MCI would satisfy a diagnosis of *probable* pDLB. The current study predicts that due to the fact that their supportive biomarker (RSWA) was not orthogonal to the core feature (RBD), *probable* pDLB patients classified using these criteria will be less likely to transition to DLB than probable pDLB patients classified with a supportive biomarker other than RSWA). Furthermore, *probable* pDLB patients who have been classified with a supportive biomarker that is not RWSA will also perform poorly at the SART, MR and BPP tasks, further supporting their risk of transitioning to DLB.

## 2. Methods

### 2.1. Participants

A total of 47 iRBD participants, 6 female, with a mean age of 68.7 (SD = 8.0) years, were recruited from the ForeFront Parkinson’s Disease Research Clinic at the Brain and Mind Centre, University of Sydney, where they form part of an ongoing, prospective longitudinal study that typically aims to review patients every 12 months. Unfortunately, the COVID-19 pandemic impacted upon this longitudinal review, but efforts to confirm whether subjects had remained clinically stable or had transitioned to a synucleinopathy were pursued via several avenues, including in-person outpatient clinical review and via telehealth. The study was approved by the Human Research Ethics Committee at the University of Sydney (ethics number = 2013/945), and all participants provided written informed consent.

A diagnosis of iRBD was confirmed prior to recruitment according to the International Classification of Sleep Disorders, 3rd Edition [41], where the electromyographic recordings demonstrating RSWA on vPSG were recorded utilising the SINBAR protocol [47,48]. Whilst full (research-focused) follow-up assessments would typically occur every 12 months (or sooner, depending on the emergence of symptoms), the impact of the COVID-19 pandemic meant that many patients were seen for clinical purposes in person or via telehealth. The restricted access and health concerns resulting from the pandemic resulted in a range of follow-up intervals ranging from 17 to 47 months. At their follow-up assessment, patients were classified as transitioned if they satisfied the Movement Disorder Society’s (MDS) criteria for PD [36] or the International DLB Consortium criteria for DLB [10].

### 2.2. Clinical Assessments

At baseline, all clinical assessments were undertaken by a qualified clinician and participants were assessed for the presence of other neurodegenerative, neurological, psychiatric or general medical conditions that would impact on their diagnosis/assessment. In addition, they underwent a semi-structured interview, which screened for salient prodromal core features of CF and VH. Visual hallucinations were also objectively assessed using item 1.2 from the MDS Unified Parkinson’s Disease Rating Scale (MDS-UPDRS), which uses a five-point Likert scale ranging from 0 (‘‘no hallucinations or psychotic behaviour”) to 4 (‘‘patient has delusions and paranoia”) [35], as well as the PsycH-Q, a self-report questionnaire that measures the characteristics of hallucinations in addition to the severity of these symptoms [49]. The core feature of parkinsonism was objectively measured using section three of the MDS-UPDRS [35] and a trained movement disorders neurologist evaluated whether they met clinical criteria for a diagnosis of PD evaluating features such as bradykinesia with decrement, rigidity with cogwheeling and the nature of any tremor. The RBD screening questionnaire was used to help quantify the degree of dream enactment experienced [50]. Given their highlighted role in predicting synucleinopathy, colour discrimination and contrast sensitivity were assessed using the Farnsworth–Munsell colour test [51] and Pelli–Robson chart [52], respectively, whilst olfaction was measured using Sniffin Sticks testing [53].

Participants were tested on a cognitive battery including the Montreal Cognitive Assessment (MoCA) [54] as a measure of global cognition; memory was assessed using the Logical Memory 1 and 2 tasks from the Wechsler Memory Scale-Revised [55] and Rey Adult Verbal Learning task [56]; executive function was assessed using the forward and backward Digit Span [57] task to measure working memory [58]; visuospatial ability was assessed with the TMT-A [58], copying a wire cube and the clock drawing task [59]; attention was assessed with the TMT-B [58] and parts 3 and 4 of the Stroop [58]; language was assessed with the COWAT word fluency for the letters (F, A, S) and categorical fluency for naming animals [57]. More specifically, a diagnosis of mild cognitive impairment was defined according to the International Working Group on MCI criteria [60,61], which specifies a complaint of cognitive impairment from the patient or informant, which is supported by impairment in two cognitive domains. Impairment within a cognitive domain was determined by isolating components of the MoCA for each cognitive domain and applying impairment thresholds as outlined in Szeto et al. [62]. Memory was assessed with the word recall component, where less than three recalled words indicated impaired memory. Executive function was assessed with Digit Span backwards where a score of zero was rated as impaired. Visuospatial impairment was assessed with the cube copy item, where a score of zero indicated impairment [54]. Impairment within the attention domain was assessed using “serial 7s” with a score less than three correct subtractions representing impairment [54]. Language was assessed with the verbal fluency task, where less than 12 words generated in 60 s was scored as impaired [54]. Based on these assessments the iRBD patients were then grouped into patients who were cognitively normal (iRBD-CN) and patients with iRBD-MCI who were regarded by the current research criteria as having *probable* prodromal DLB. However, *probable* prodromal DLB was also assessed ignoring the presence of RWSA as a supportive biomarker and such patients were required to have at least one additional core feature from cognitive fluctuations (CF), visual hallucinations (VH) and/or parkinsonism that did not meet diagnostic criteria for PD. Other biomarkers (DaTSCAN, MIBG) were not available in these patients.

### 2.3. Novel Tasks for Assessing Prodromal DLB

The SART is a task of attention that probes the core feature of cognitive fluctuations (CF) by presenting the subject with a series of numbers between 1 and 9. Patients are required to press a button when any number is presented but withhold their response when the number “3” is displayed (for full description, see Phillips et al. [43]). The number of missed responses and variance in response time (RTSD) have been shown to correlate with CF in DLB patients [43].

The MR task measures visuospatial impairment by presenting subjects with pairs of shapes. Patients were required to indicate if the shapes were identical or a mirror image of each other. Trials may involve shapes facing the same direction or where one shape is rotated either to 45° or 90° [43]. Accuracy is the main outcome of the MR, with the lowest accuracy limited to 50%, representing chance performance. Accuracy on the MR task has been shown to be correlated with VH and visuospatial ability in DLB patients [43].

The BPP was used to provide an objective measure of VH. In this task, patients are presented with ambiguous images that contain either one (single image) or two percepts (hidden or bistable). Patients are then required to indicate if there are one or two percepts in each image being displayed. Percepts that were misidentified were labelled as misperceptions, whilst percepts that were not identified were labelled as misses (Shine et al., 2012; for a full description, see Phillips et al. [46]). The BPP has been shown to correlate with VH in patients with PD [45] and DLB [46].

Unfortunately, the time available to administer the SART, MR and BPP was limited in those participants recruited in the later stages of the study, which was impacted by testing restrictions imposed during the COVID-19 pandemic. This resulted in different numbers of iRBD patients being tested across the SART, MR and BPP paradigms (see Table 1). These restrictions also hindered our ability to repeat assessments at annual follow-up, but every effort was made to at least contact the patient to determine if they had transitioned from iRBD to a synucleinopathy.

### 2.4. Statistics

All analysis was performed using SPSS version 26 [63]. Demographic measures across the iRBD-CN and iRBD-MCI groups were not normally distributed, and as such, group comparisons were made using Mann–Whitney U tests. Furthermore, the number of misses, false alarms and the standard deviation of response time from the SART and accuracy performance on the MR task were not normally distributed; thus, between-group comparisons were performed with a Mann–Whitney U test. Further analysis to control for any significant covariates was performed with an ANCOVA. The low number of iRBD patients completing the BPP restricted any between group comparisons, but the effects of any significant covariates were assessed using a hierarchical, multiple regression. Due to the low number of patients who transitioned to DLB or PD, baseline performance on the SART, MR and BPP were simply rank ordered across all participants to assess whether these individuals showed any impairments that might have been indicative of their likelihood for transition. The most impaired score of each measure was ranked “1”.

## 3. Results

### 3.1. Demographics

Demographics are presented in Table 1. Mild cognitive impairment was detected in 13 of the 47 iRBD patients. Within these 13 patients, clinical interview failed to detect any CF or VH. Across all the iRBD patients, the MDS-UPDRS III scores ranged from 0 to 18, but none satisfied clinical criteria for a diagnosis of PD. However, one iRBD patient with MCI also demonstrated clinical parkinsonism having unilateral bradykinesia with decrement. Thus, 12 of the 13 patients in the iRBD-MCI group were stratified as having *probable* prodromal DLB on the currently proposed research criteria, but with confounded clinical (RBD) and subjective biomarker (RSWA). Thus, only one iRBD-MCI satisfied *probable* MCI-LB using orthogonal measures, where two core features (RBD and clinical parkinsonism) were present in addition to RSWA. For group comparisons, all iRBD-MCI patients were combined into one group for comparisons with the iRBD-CN group. The iRBD-CN and iRBD-MCI groups were matched for age and RBD symptom duration. However, the iRBD-CN patients had more years of education than the iRBD-MCI patients that conducted each of the SART (z(*n* = 31) = −2.39, *p* = 0.02, r^2^ = 0.18), MR (z(*n* = 46) = −2.36, *p* < 0.01, r^2^ = 0.12) and BPP (z(*n* = 23) = −3.00, *p* < 0.01, r^2^ = 0.39).

Over the period of follow-up (range 17–45 months), six iRBD patients were able to undergo full follow-up assessment, 20 patients were clinically assessed at follow-up but did not undergo quantitative battery (5 patients were assessed via telehealth). Twenty-one patients were unable to attend follow-up appointments due to COVID restrictions, change of contact details or refusal to continue testing.

From the follow-ups, two iRBD patients had transitioned to a synucleinopathy. One iRBD-CN patient was diagnosed with PD (33 months from their baseline visit) and one iRBD-MCI patient had transitioned to DLB (28 months from their baseline visit), see Table 2. Interestingly, this iRBD-MCI patient was the only one who had two core features (RBD and clinical parkinsonism), plus a supportive biomarker (RSWA) at baseline.

### 3.2. Task Performance

#### 3.2.1. Sustained Attention Response Task

The iRBD-CN and iRBD-MCI groups had similar performance on the SART with no difference between misses (z(*n* = 31) = −1.65, *p* = 0.16, r^2^ = 0.09), false alarms (z(*n* = 31) = −0.59, *p* = 0.58, r^2^ = 0.01) or variance in response time (z(*n* = 31) = −0.78, *p* = 0.45, r^2^ = 0.02; see Table 1; Appendix A). No further analysis was performed.

#### 3.2.2. Mental Rotation

The iRBD-CN group had higher accuracy at the 0°, 45°, 90° (Appendix A) conditions compared to the iRBD-MCI group (0°: z(*n* = 46) = −2.74, *p* = 0.01, r^2^ = 0.16; 45°: z(*n* = 46) = −2.11, *p* = 0.04, r^2^ = 0.10; 90°: z(*n* = 46) = −2.26, *p* = 0.03, r^2^ = 0.10) and this was also reflected in the overall accuracy (z(*n* = 46) = −2.39, *p* = 0.02, r^2^ = 0.12). However, this group effect was removed across each condition after controlling for years of education with an ANOVA (0°: F(1, 43) = 2.67, *p* = 0.11, η^2^ = 0.06, β = 0.36; 45°: F(1, 43) = 2.78, *p* = 0.10, η^2^ = 0.06, β = 0.37; 90°: F(1, 43) = 2.61, *p* = 0.11, η^2^ = 0.06, β = 0.35). Whilst not significant, the accuracy on the MR did appear to be trending towards a significant difference (F(1, 43) = 3.64, *p* = 0.06, η^2^ = 0.09, β = 0.46) between the groups despite controlling for years of education, which might reflect an under-powered study.

#### 3.2.3. Bistable Percept Paradigm

The group sizes for the BPP task were very unbalanced, with only four iRBD-MCI participants tested (see Table 1; Appendix A), and whilst there was a significant difference with iRBD-MCI patients missing more percepts than iRBD-CN, (z(*n* = 23) = −2.47, *p* = 0.01, r^2^ = 0.26), there were no differences between misperceptions (z(*n* = 23) = −1.44, *p* = 0.15, r^2^ = 0.09) or the BPP error rate (z(*n* = 23) = −1.73, *p* = 0.08, r^2^ = 0.13). The groups were also unmatched for years of education and a small sample size restricted further analysis of this effect.

### 3.3. Transitioned Patients

During the period of the study, two patients transitioned, one to DLB and one to PD. The transitioned DLB patient was male, 84 years old and diagnosed with iRBD 7 years prior to their DLB diagnosis (see Table 2). On their first assessment, they scored poorly on the MoCA (23/30), indicating moderate cognitive impairment, but were not demented. At that assessment, there were no CF or VH and whilst they had parkinsonism (bradykinesia with decrement), they did not satisfy diagnostic criteria for PD. Thus, at their baseline assessment, they satisfied the research criteria for *probable* MCI-LB and transitioned to DLB after a further 28 months. They performed well on the SART with no false alarms or misses, and across the 31 patients tested, they ranked 9 out of 31 for RTSD, suggesting high variance in response time. Their accuracy was varied on the MR task. Out of the 46 patients tested, they ranked 28th out of 31 with an accuracy of 72% at 0° rotation, the second lowest score of 61% at 45°, but at 90° they performed relatively well, scoring 78%, which was ranked 21st. Overall, on the MR they ranked 24 out of 31 with an accuracy of 70.4%, which was the seventh-lowest accuracy and lower than the mean accuracies of the iRBD-CN and iRBD-MCI groups. Their performance on the BPP was also significantly impaired, ranked 2 out of 31 for the number of misses, although they had no misperceptions.

The patient who transitioned to PD was a 65-year-old male who was diagnosed with iRBD for 5 years prior to their clinical diagnosis. They scored well on the MoCA (28/30) and did not meet criteria for MCI (see Table 2). They also performed well on the SART with no misses or false alarms and had a consistent response time. For the MR task, they performed well at 0°, but poorly for the 45° condition, with the second-lowest accuracy out of the 46 patients. They had moderate performance impairment on the 90° condition scoring 72.0% and had the tenth-lowest accuracy overall of 74.1%. Their BPP score was within the normal range with an error score of 7.5%, which is below the threshold reported for impairment (11%; Shine et al., 2012). Clinically, they did have mild parkinsonism that did not reach diagnostic criteria for PD at baseline assessment with only mild bradykinesia. Obviously, the absence of MCI precluded a diagnosis of MCI-LB in this patient.

## 4. Discussion

The aim of the current study was to test the performance of potential prodromal DLB patients on the SART, MR and BPP and to capture the baseline performance on these tasks in patients that later transitioned to DLB or PD. The data revealed no striking performance differences between patients with iRBD-CN and iRBD-MCI, but the study was impacted by low sample sizes as a result of the restrictions imposed by the COVID-19 pandemic. However, this study did highlight the value of classifying probable prodromal DLB using orthogonal clinical and objective biomarker measures. Of the 13 patients with PSG-confirmed iRBD who demonstrated MCI, only one had an additional core clinical feature that allowed them to be classified as having *probable* prodromal DLB, ignoring the presence of RSWA, and it was this individual patient who transitioned to DLB.

No significant differences were observed between the iRBD-CN and iRBD-MCI groups on measures of the SART with a generally low number of misses and false alarms, along with having a consistent response time. These findings support a previous study that correlated SART performance to CF severity in DLB patients [43]. As CF was not reported within this iRBD cohort, it was only to be expected that both groups would perform similarly. Findings from the current study also support previous work indicating that performance on the SART is not affected by the degree of global cognitive impairment in DLB [43].

Impairments in attention are a characteristic finding in DLB, and some previous studies have identified this in MCI-LB patients using Digit Span, TMT part B and Stroop [64,65,66,67]. These conflicting findings may be due to several reasons, including the fact that previous studies investigated MCI-LB patients that were diagnosed by combining MCI, DLB or PDD criteria [65,66] or by retrospectively assigning a diagnosis depending on the individual’s clinical transition [64]. These studies also reported higher frequencies of CF, VH and parkinsonism, suggesting a more severe population than tested in the current study. Furthermore, tasks used by previous studies may rely on a different mechanism than the SART, such as attentional switching or shifting. Indeed, the tasks used by previous studies required shorter attention spans than the SART, suggesting different mechanisms are required to sustain attention compared to those that rely on short bursts of attention.

An initial analysis highlighted lower accuracy on the MR task amongst the iRBD-MCI participants, but this effect may have been driven by years of education. Greater years of education have been found to preserve cognition and delay cognitive decline in older populations [68,69]. The lower performance in the MR may be attributed to the greater number of years the iRBD-CN had spent in education compared to the iRBD-MCI group, rather than the disease itself. However, the group effect was trending towards significance for overall accuracy (F(1, 43) = 3.64, *p* = 0.06, η^2^ = 0.09, β = 0.46) and the low beta value would indicate that this comparison was under-powered. Thus, it is possible that a larger sample size might be required to further evaluate this potential relationship. Previous findings have suggested that the MR task is reliant on visuospatial ability in DLB patients [44], which might suggest that visuospatial impairment occurs very early in DLB, before other core features emerge. An interesting additional observation was the number of patients that had low accuracy at the 0° condition from both groups. This condition involves a straight comparison of two relatively simple shapes, and it was surprising that six of the iRBD-CN patients did not perform this task with near-perfect accuracy (>90%). However, performance on the MR has been found to be unrelated to the level of cognitive impairment per se [44], suggesting that performance at the 0° condition may reflect a selective, sub-clinical visuospatial impairment in both groups.

Whilst the iRBD-MCI group appeared to perform more poorly on the BPP than the iRBD-CN group, this result should be interpreted with caution, due to the small sample size and the difference in years of education between the groups in favour of the iRBD-CN patients. It should also be noted that previous studies have found the BPP to be correlated with self-reported VH in PD [45] and DLB [46], whereas the iRBD patients in the current study had no reports of these phenomena. However, a higher number of misses by iRBD-MCI patients would be in keeping with previous work that reported DLB patients had more misses than misperceptions on the BPP [46], a finding that may reflect impaired attentional switching or reduced saccadic processes in DLB patients [46].

In the current study, two patients who were assessed at baseline transitioned to a clinical synucleinopathy at follow-up over a minimum period of 28 months. At their baseline assessment, the patient transitioning to DLB was the only participant classified as meeting criteria for *probable* prodromal DLB by the presence of two core clinical features (RBD and parkinsonism), as well as RSWA on their PSG. However, they did not endorse cognitive fluctuations or visual hallucinations at this visit. In keeping with the absence of cognitive fluctuations, they performed relatively well on the SART but their performance on the MR task was variable, with poor performance at the 0 and 45° conditions but reasonable with the 90° trials, which might suggest latent CF. Finally, performance on the BPP of the patient who transitioned to DLB resembled what has been reported in previous work in DLB patients who experience VH, where subjects have a higher rate of trials where they fail to identify ‘hidden’ percepts [46]. The variable pattern of performance across the SART, MR and BPP in this individual who transitioned to DLB highlights the heterogeneity that exists not only across patients with DLB, but also within individuals who may have a differential pattern of impairments across attention and visuoperceptive function. In contrast, the patient who transitioned to PD was younger (only 68 years of age) at baseline with a five-year history of iRBD and did not have MCI. They performed very well at the SART and did not reach the threshold to suggest impairment on the BPP. Their performance on the MR was mixed, suggesting potentially selective visuospatial impairment.

There are several significant limitations in the current study that need to be considered when interpreting the findings. The first relates to the small number of patients that had transitioned during the average follow-up period of 31 months. The transition rate of iRBD patients into either PD or DLB has been reported as being around 6% per annum [20] and 35% at five years [29], but results have varied from 6% to 10% per annum across studies [20,70,71,72]. From the current cohort of 47 patients, this would have predicted 7–12 patients over the period evaluated. Unfortunately, not all patients were able to attend a full in-person follow-up due to COVID-19 restrictions, meaning that some subtle features that might have permitted a clinical diagnosis may have gone undetected. In addition, 21 patients declined or were lost to follow-up, which might reasonably have included those who had transitioned. Extending the follow-up period may have also been beneficial, as it would have increased the chances of iRBD patients transitioning to PD or DLB.

Due to the cognitive battery applied, a more detailed classification of MCI was not attempted. However, the current study did identify patients with two or more domain impairments as MCI where the domains were taken from relevant sections of the MoCA with thresholds drawn from previously published work [42], noting that a diagnosis of MCI cannot be made using MoCA alone [73]. This method of defining MCI patients reflects the current MCI guidelines that suggest test scores that are 1–1.5 standard deviations below normal to be considered as impaired [60]. Previous studies have found that MCI patients with only amnestic impairments are more likely to transition to AD, whereas MCI patients with multiple domain impairments or only non-amnestic domains are more likely to transition to DLB [60]. Making this distinction in the current study may have yielded stronger effects if MCI with only amnestic impairments were excluded. However, it should be emphasised that the presence of iRBD is extremely rare in AD [74]. This suggests that if patients within the current cohort were to transition, it would likely be to a synucleinopathy. 

It should also be noted that while none of our patients reported VH or CF, the BPP and SART still provide utility, as they are designed to measure subclinical features of VH, through misperceptions and misses, while the SART may be sensitive to fluctuations of much lower amplitude. While patients may not indicate that they experience either VH or CF, if they do have pDLB, these symptoms are likely to be present, just in a very mild form. Further longitudinal studies are needed to investigate how sensitive these tasks are to very mild forms of VH and CF.

Finally, the approach taken here stipulated that MCI patients with PSG-confirmed RBD should be regarded as meeting a diagnosis for only *possible* prodromal DLB, which was the situation for 13 participants. Only one MCI patient who had PSG-confirmed RBD also had a second core feature (parkinsonism), which resulted in their classification as *probable* prodromal DLB. Significantly, this was the only patient who transitioned to DLB, emphasizing the importance of employing orthogonal clinical core features and objective biomarkers.

## 5. Conclusions

In summary, the results of the current study would support the need for more stringent criteria to allow a more accurate stratification for the risk of developing DLB. This approach would demand that orthogonal measures should be adopted rather than accepting the circularity of combining the presence of clinical RBD and RSWA in a patient with MCI, as reflecting *probable* MCI-LB. We would propose that the currently proposed research criteria for MCI-LB should specify that the combination of clinically reported RBD (one core feature) that is confirmed by RSWA (one objective biomarker) in patients with MCI should be insufficient to meet a diagnosis of *probable* prodromal DLB. Previous studies have found utility of the SART, MR and BPP in detecting DLB core symptoms [43,44,46]; findings are less clear for patients in the prodromal stages of this disease. It is clear that more systematized work with larger cohorts and longitudinal assessment is required to confirm their future role.

## Figures and Tables

**Table 1 brainsci-12-01594-t001:** Demographic and task performance.

	iRBD-CN Mean (SD)	iRBD-MCI Mean (SD)
Total N	34	13
**Sustained Attention Response Task**		
N	20	11
Age	68.5 (7.9)	70.2 (9.3)
Gender m(f)	17 (3)	8 (3)
Disease duration	1.8 (2.8)	2.9 (4.2)
Years of education	15.2 (2.2)	12.9 (2.7) ^a^
MoCA	27.6 (2.0)	23.6 (2.5) ^a^
MDS-UPDRS III	9.8 (7.1)	8.1 (5.3)
Misses	1.8 (4.1)	1.3 (1.5)
False alarms	2.9 (3.3)	3.2 (3.1)
RTSD	107.33 (80.21)	92.80 (28.97)
**Mental Rotation**		
N	34	12
Age	67.9 (6.9)	70.1 (8.9)
Gender	19 (5)	9 (3)
Duration	2.8 (5.4)	2.7 (4.1)
Years of education	15.1 (2.7)	12.9 (2.6) ^a^
MoCA	27.9 (1.7)	23.3 (2.7) ^a^
MDS-UPDRS III	9.5 (6.5)	8.9 (5.8)
0°	94.9% (9.1%)	86.0% (13.6%)
45°	87.5% (11.6%)	77.6% (15.4%)
90°	79.4% (14.4%)	67.2% (16.3%)
Total	87.3% (10.2%)	77.0% (12.9%)
**Bistable Percept Paradigm**		
N	19	4
Age	67.6 (8.9)	67.2 (7.0)
Gender	17 (2)	3 (1)
Duration	1.5 (2.7)	0.5 (0.57)
Years of education	15.3 (2.2)	10 (1.6) ^a^
MoCA	27.1 (2.6)	24.3 (2.6) ^a^
MDS-UPDRS III	9.5 (7.1)	7.3 (3.8)
Misses	4.9 (2.3)	8.8 (4.0) ^a^
Misperceptions	2.4 (2.2)	1.25 (1.3)
Error score	9.1% (3.7%)	12.6% (4.4%)

iRBD-CN = cognitive normal iRBD; MoCA = Montreal Cognitive Assessment; MDS-UPDRS = Movement Disorder Society Unified Parkinson’s Disease Rating Scale; RTSD = response time standard deviation; age, disease duration and years of education are presented in years. ^a^ = *p* < 0.05.

**Table 2 brainsci-12-01594-t002:** Baseline demographic and task performance between transition and stable iRBD patients. Rank of 1 indicates the most impaired of the cohort.

	DLB (Rank; Total)	PD (Rank; Total)	iRBD-CN Mean(SD)	iRBD-MCI Mean(SD)
Total N	1	1	34	12
Age	84	68	69.3(7.4)	70.5(7.7)
Gender m(f)	Male	Male	36(9)	12(3)
Disease duration	-	-	3.2(5.0)	2.7(3.8)
Years to transition	7	5	-	-
Years of education	17	13	14.5(3.2)	12.3(2.7)
MoCA	23	28	26.8(4.5)	23.5(2.3)
MDS-UPDRS III	9	12	9.9(8.2)	9.5(6.6)
SART				
Misses	0(13; 31)	0(13; 31)	1.8(4.2)	1.4(1.5)
False alarms	0(27; 31)	0(27; 31)	3.3(3.5)	3.5(3.0)
RTSD	111.6(9; 31)	56.3 (29; 31)	110.0(90.9)	90.9(29.8)
Mental Rotation				
0°	72.0%(4; 46)	89.0%(9; 46)	95.0%(9.2%)	87.3%(13.5%)
45°	61.0%(2; 46)	61.0%(2; 46)	88.3%(10.7%)	79.1%(15.2%)
90°	78.0%(21; 46)	72.0%(19; 46)	80.0%(14.6%)	66.2%(16.7%)
Total	70.4%(7; 46)	74.1%(10; 46)	87.7%(10.1%)	77.6%(13.4%)
BPP				
Misses	11(2; 23)	6(7; 23)	4.5(1.9)	8.6(4.0)
Misperceptions	0(18; 23)	0(18; 23)	2.6(2.1)	1.3(1.3)
Error	13.8%(2; 23)	7.5%(7; 23)	9.0%(3.7%)	12.5%(4.5%)

PD = Parkinson’s disease; DLB = dementia with Lewy bodies; iRBD-CN = cognitively normal iRBD; iRBD-MCI = mild cognitive impairment; MoCA = Montreal Cognitive Assessment; MDS-UPDRS = Movement Disorder Society Unified Parkinson’s Disease Rating Scale; SART = Sustained Attention Response Task; RTSD = response time standard deviation; BPP = Bistable Percept Paradigm; age, disease duration, year to transition and years of education are presented in years.

## Data Availability

Data is available upon request.

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
