# Peer review of "Exploring the Sensitivity of Prodromal Dementia with Lewy Bodies Research Criteria"

_brainsci, 2022, doi:10.3390/brainsci12121594_

Round 1
Reviewer 1 Report
Summary:
The study by Phillips et al evaluated the probability of transition of patients from isolated Rapid Eye Movement (REM) Sleep Behavior Disorder( iRBD) to PD/DLB over the course of the study. To address this, they recruited 47 iRBD patients after their diagnosis was confirmed on vPSG. The authors found 2 out of 47 iRBD patients transitioned to synucleinopathy (one PD and one DLB). satisfied the research criteria for probable MCI-LB and transitioned to DLB after a further 28 months. The patient that transitioned to PD did not reach the diagnostic criteria for PD at the baseline assessment however, the patient had mild parkinsonism at the time of assessment. The authors suggested having stringent.
They also evaluated 3 novel behavior assessments in these iRBD patients; sustained Attention Response Task (SART), the Mental Rotation (MR), and the Bistable Percept Paradigm(BPP) in a cohort of iRBD to assess their predictive capacity for prodromal DLB cases. They divided the iRBD cohort into two subgroups which were cognitively normal (iRBD-CN) and others that had MCI (iRBD-MCI). They did not find any stark difference in any of these tests between the iRBD-CN and iRBD-MCI cohorts.
Overall, the authors have performed a detailed assessment of the patients along the lines of the current research criteria. The authors have also mentioned the low sample size and not being able to follow up on all the patients due to COVID-19 restrictions led to this study being underpowered to draw much significance from the data.
General Comments:
1) The introduction needs to be rearranged to clearly define what is previously known facts and what are the exact questions the authors want to address in this study and why. The same should also be reflected in the discussion section where they could emphasize more the key objectives addressed in the manuscript.
2) Since there are a lot of abbreviations used, a table listing these abbreviations would be very helpful to the readers.
3) A short description of the common terminologies used in the paper like Prodromal DLB, cognitive fluctuations (CF), or metaiodobenzylguanidine (MIBG) would help the reader better understand the study.
4) Since the authors discuss the current research criteria and diagnostic criteria in the manuscript, it would be great if the authors could include a table with a summary of the current criteria as supplementary information with appropriate citations.
5) Since the manuscript is focused on iRBD patients, it would be good if authors could also describe this group of patients in a little brief in the neurodegeneration field.
6) The authors mention a battery of three tests SART, MR, and BPP which they used for the assessment of the iRBD patients. It would be great if the authors could discuss the importance of these tests in the introduction section regarding the neurodegeneration landscape specifically PD/DLB based on prior research done in the field.
7) Line 453: Seems like the sentence ’The results presented here’ has been typed by mistake.
Specific Comments
1) In table 2, since there have been different numbers of patient participants in the battery of tests used, it would be better to represent the rank with the total participant together to better understand the strength of the test.
2) The authors included the battery of tests in which they mention that the accuracy of MR and BPP is correlated with VH. Since none of the patients had VH or CH, no stark difference was found among the groups compared. Also, none of the patients including the transitioned PD and DLB patients had VH or CH. A short explanation in the discussion section on how to interpret these data in terms of usefulness in iRDB patients with no VH and CH would be beneficial.
3) The authors claim that the group effect in the MR score was insignificant after controlling for years of education. It would be nice to support this statement with some references to better understand how education affects these scores.
4) The authors claimed that the transitioned DLB patient showed reasonable performance at 90 degrees could be due to latent CF. Did the authors perform these tests after the patient's transition to see a worsening of performance in this test?
5) In the conclusion section, the authors mention the potential utility of SART, MR, and BPP as tools for the assessment of patients with prodromal DLB, however, the data presented in the manuscript is not enough to make this claim considering these tests correlate with visual and cognitive hallucinations and none of the patient cohorts had VH or CH.
Reviewer 2 Report
The aim of the current study was to test the performance of potential prodromal DLB patients on the SART, MR and BPP and to capture the baseline performance on these tasks in patients that later transitioned to DLB or PD.
Unfortunately, this manuscript needs substantial improvements and corrections before publishing may be possible.
General points:
Please add a list of abbreviations before References section to your manuscript.
Please do your list of references according to” Brain Sciences”.
Special points:
Keywords: isolated REM sleep behaviour disorder (iRBD)
Introduction
Lines 45-56: please add multiple or more references at the end of each these sentences.
Lines 60-63: please add multiple or more references at the end of each these sentences.
Lines 100-102: please add multiple or more references at the end of each these sentences.
Methods
Please add the exactly information about all your patients: number totally, number of female, number of male and the mean age.
Lines 137-138: please add also the exactly organisation name, date and the number of the permission for all your experiments.
Lines 160-162: please add multiple references at the end of this sentence.
Lines 194-206: please add multiple or more references at the end of each these sentences.
Lines 236-248: please add also the exactly software version number.
Results
For better readability, please add also some diagrams to your results.
Round 2
Reviewer 2 Report
Dear authors,
Thank you for your corrections.
Unfortunately, this manuscript needs once more improvements and corrections before publishing may be possible.
Please add a Future perspectives section to your manuscript.
Please check and correct all spaces between the references numbers at the end of each sentences and the words.
Introduction
Lines 47-127: please add multiple or more references at the end of each these sentences.
Supplement Figures
Please add a Legends with exactly description for each Supplement Figures.
Author Response
Thank you for your suggestions. We have updated the manuscript according to these comments, with responses below:
Please add a Future perspectives section to your manuscript.
Future perspectives are included in the conclusion section of the manuscript. This section states that the current diagnostic criteria should be changed to require iRBD to only be included as a clinical OR biomarker criteria when classifying potential pDLB patients.
Lines 47-127: please add multiple or more references at the end of each these sentences.
Noting the reviewers' comments, we have added multiple references support the appropriate sentences in the introduction.